# An Immature Dermapteran Misidentified as an Adult Zorapteran: The Case of *Formosozoros newi* Chao & Chen, 2000

**DOI:** 10.3390/insects14010053

**Published:** 2023-01-05

**Authors:** Petr Kočárek, Fang-Shuo Hu

**Affiliations:** 1Department of Biology and Ecology, University of Ostrava, Chittussiho 10, CZ-71000 Ostrava, Czech Republic; 2Department of Biological Sciences, National Sun Yat-sen University, 70 Lienhai Rd., Kaohsiung 80424, Taiwan

**Keywords:** Polyneoptera, *Zorotypus*, taxonomy, classification, Taiwan, Oriental region

## Abstract

**Simple Summary:**

The paper corrects a mistake in the identification of *Formosozoros newi* Chao & Chen, 2000, described in Zoraptera. *F. newi* is characterized by extremely long cerci and tarsi with a long first tarsomere; the latter character has not been recognized in any other extant or fossil Zoraptera. Based on the morphological comparisons of diagnostic characters, we here identified *Formosozoros newi* as the 1st-instar nymph of an earwig (Dermaptera).

**Abstract:**

Zoraptera shows extreme uniformity in general body morphology, with the exception of *Formosozoros* (=*Zorotypus*) *newi* Chao & Chen, 2000, which stands out in terms of the shape and arrangement of its legs, the cerci, and several other morphological characters. After critical evaluation, we found that this species is not a zorapteran but is instead a nymph (1st instar) of an earwig; i.e., *F. newi* is a dermapteran. Because of the lack of morphological descriptions of Dermaptera nymphs that would allow species identification, and because the type material of *F. newi* is lost, a more detailed classification is not possible. We therefore propose that the genus name *Formosozoros* Chao & Chen, 2000 and the species name *Formosozoros newi* Chao & Chen, 2000 are nomina dubia.

## 1. Introduction

Zoraptera are small (2–3 mm), inconspicuous insects distributed mainly in tropical regions [1,2]. The extant diversity of Zoraptera is low, with only 43 described species [3,4]. All members of the order Zoraptera show extreme uniformity in general body morphology, and this has led to the persistence of a conservative classification of Zoraptera with only one nominotypic genus (*Zorotypus* Silvestri, 1913) in one family (Zorotypidae) for >100 years, i.e., until 2020 [3,5]. Kukalova-Peck and Peck (1993) [6] divided the Zoraptera into seven genera based on wing venation. Engel and Grimaldi (2000) [7], however, critically revised the supraspecific classification and concluded that the proposed generic characters concerning wing venation are either continuous across taxa or variable within a given species. Only the use of molecular methods helped to clarify the systematics of Zoraptera. Matsumura et al. (2020) and Kočárek et al. (2020) [3,8] independently conducted molecular phylogenetic studies using a combination of nuclear and mitochondrial markers, and both analyses revealed two major phylogenetic lineages, which were classified by Kočárek et al. (2020) [3] as families (Zorotypidae Silvestri, 1913 and Spiralizoridae Kočárek, Horká & Kundrata, 2020). Each of the families was divided into two well-supported subclades, i.e., subfamilies with 9 valid, currently recognized genera [3]. The genera are mainly diagnosed based on male genitalia, abdominal sclerites, and hind legs.

Chao and Chen (2000) [9] described a new genus, *Formosozoros* Chao & Chen, 2000, that was based on a single species, *Formosozoros newi* Chao & Chen, 2000, collected from Taiwan. The type material of this species consisted of five females. The genus was characterized by extremely long cerci and tarsi with a long first tarsomere; the latter character has not been recognized in any other extant or fossil Zoraptera. Nevertheless, Engel and Grimaldi (2000) [7] immediately synonymized the genus *Formosozoros* with Zo*rotypus* Silvestri, 1913. Because males were unknown and type material was unavailable, Kočárek et al. (2020) [3] did not assign *F. newi* to the newly proposed classification and considered the species as incertae sedis in Zoraptera. 

In this report, we describe the morphology of *F. newi* and critically evaluate its diagnostic characters, thereby clarifying its taxonomical status.

## 2. Materials and Methods

We compared the original description and drawings of *F. newi* (Figures 1A and 2A,D,G) with voucher specimens of extant Zoraptera from all currently valid subfamilies (for a list, see Kočárek et al. (2020) [3]) and describe how *F. newi* differs morphologically from a representative of Spiralizoridae: Spiralizorinae, the most widespread Zoraptera group in SE Asia [3]. To compare *F. newi* with a dermapteran, we used the visually similar nymphs of *Paralabellula curvicauda* (Motschulsky, 1863) (Dermaptera: Spongiphoridae). The latter nymphs were obtained from a breeding culture established with adults collected in the same habitat as reported for *F. newi* in Yuemei, Yilan County, Taiwan (Figure 1C,D). The adults were placed in a plastic container along with the pieces of dead wood from the collecting habitat, and no other food resources were provided. The 1st-instar nymphs were found after a few months. These and more mature nymphs were stored in 96% ethanol. All specimens were studied and photographed with a Leica Z16 APO macroscope equipped with a Canon 6D Mark II camera and with an Olympus CX41 microscope equipped with a Canon D1000 camera. Micrographs of 20 to 30 focal layers of the same specimen were combined with Helicon Focus software and finally processed with Adobe Photoshop CS6 Extended (version 13). Living individuals were photographed with Canon EOS R5 and Canon RF 100 mm f/2.8 L Macro IS USM lenses.

The morphological description of *F. newi* presented in the results was extracted from Chao and Chen (2000) [9], and illustrations of *F. newi* were redrawn from the original drawings of Chao and Chen (2000) [9]. Figures are reproduced with permission from journal *Pan-*Pacific Entomologist** published by the Pacific Coast Entomological Society (PCES) in 2000.

Material studied: *Spiralizoros cervicornis* (Mashimo, Yoshizawa & Engel, 2013)—Brunei Darussalam, Ulu Temburong NP, Sungai Esu stream, GPS: 04°32′14.1″ N, 115° 9′47.1″ E, 150 m a.s.l., 9.i.2014, P. Kočárek leg.; *Paralabellula curvicauda* (Motschulsky, 1863)—Taiwan: Yilan, Luodong To., 24°40′22.1″ N, 121°46′58.4″ E, i.–ii.2022, leg. F.S. Hu and M. Fikáček.

Voucher specimens of *P. curvicauda* and *S. cervicornis* are deposited in the entomological collection of the University of Ostrava, Czech Republic. The classification and nomenclature are based on Kočárek et al. (2020) [3].

## 3. Results

### 3.1. Searching for the Types of Formosozoros newi

Chao and Chen (2000) [9] indicated that the types of *Formosozoros newi* had been deposited in the Department of Biology, Tunghai University, Taichung, Taiwan. However, the university lacks both slides and ethanol-preserved entomological collections. To our knowledge and based on a review of the literature, no one had examined the type material after the original description was published. The second author of the current report wrote to both Chao and Chen to ask about the types of *F. newi*, but one of the latter authors did not know the current depository, and the other did not reply. The second author of the current study also contacted the major entomological collection in Taiwan, i.e., the National Museum of Natural Science, Taichung, Taiwan (NMNS), but that museum also indicated that it did not receive the types of *F. newi*. Hence, the types of *F. newi* can be considered lost.

### 3.2. Morphological Characteristics and Distribution of Formosozoros newi

(Figure 1A and Figure 2A,D,G; description adopted from [9]).

Body length 2.88–3.14 mm. Head pear-shaped; eyes and ocelli absent; antenna 9-segmented; labial palps 3-segmented; maxillary palps 5-segmented, mandibles strongly sclerotized, almost triangular with a rounded lateral margin, with well-differentiated molar and incisor parts, incisor part with 2 teeth. Thorax segments simple, pronotum transverse; hind legs with femur expanded, with 3 posterior spines on the inner margin, tibia with row of 5 short thickened setae and 1 short apical spur, first segment of tarsus 0.18 mm in length, with row of 5–6 pairs of short thickened setae, second segment length 0.17 mm, without any thickened setae, with 2 claws. Abdominal tergite I–X with 2 medial posterior macrochaetae and 2 lateral posterior macrochaetae, tergite XI small; stemite I–IV with 4 posterior macrochaetae, stemite V-IX with 6 posterior macrochaetae, macrochaetae on stemites X–XI absent; cerci long, s-shaped, with 3–4 preapical long setae, apical seta absent.

Distribution: The species is only known from the type locality, Taiwan, Hualien.

### 3.3. Classification of Formosozoros and Formosozoros newi

Based on the evaluation of all morphological characters described and illustrated in the original descriptions of the genus *Formosozoros* Chao & Chen, 2000, and the species *Formosozoros newi* Chao & Chen, 2000, we found the following characters as diagnostic for placement of the genus and species in the order Dermaptera: mandible with two teeth in incisor part (Figure 2A), 1st tarsomere long, as long as 2nd tarsomere (Figure 2D), and cercus long and thin, s-shaped (Figure 2G).

We clearly demonstrate that the genus *Formosozoros* belongs to Dermaptera rather than Zoraptera, and we therefore reject the synonymy of *Formosozoros* Chao & Chen, 2000 with *Zorotypus* Silvestri, 1913. Because the classification of *Formosozoros* into a family/genus of Dermaptera is impossible due to the absence of detailed morphological descriptions of Dermaptera nymphs, we propose that both *Formosozoros* Chao & Chen, 2000 and *Formosozoros newi* Chao & Chen, 2000 be considered nomina dubia within Dermaptera.

## 4. Discussion

In keys used for the identification of insect orders (e.g., [10]), the most commonly used diagnostic characters for Zoraptera are unsegmented cerci, 2-segmented tarsi, and 9-segmented antennae (8-segmented in nymphs). The presence of 2-segmented tarsi is a recognized synapomorphy of Zoraptera (e.g., [2,5,11]), it is the most easily observed character, and is therefore important for the identification of Zoraptera. Three-segmented tarsi is a synapomorphy of recent earwigs (Neodermaptera), while the number of tarsal segments in fossil groups (Eodermaptera, Archidermaptera) ranges from 3 to 5. Dermaptera have multi-segmented antennae, with up to 50 segments in lower earwigs (Protodermaptera, Labiduroidea, Anisolabidoidea) and up to 23 segments in higher earwigs (Forficuloidea), whereas Spongiphoridae and Forficulidae have only 11–14 segments [12,13,14,15]. 

The segmentation of antennae and tarsi appears to be useful for distinguishing between Zoraptera and Dermaptera, because there are no overlaps in the number of segments between the two orders. These numbers, however, are based on adult earwigs. Nymphs have a smaller number of antennae segments, and 1st-instar nymphs of the most diverse Dermaptera families (Spongiphoridae, Forficulidae, and Anisolabididae) usually have only 8–9 antennal segments (e.g., [16,17,18,19]), which is the number associated with Zoraptera. Similarly, 1st-instar nymphs of Dermaptera have only 2 tarsal segments (e.g., [17]). The use of keys (e.g., [10,20]) for identification of adults to insect order can therefore lead to the misidentification of 1st-instar Dermaptera nymphs and to their erroneous classification to Zoraptera. 

One of the most important diagnostic characters for Zoraptera is the arrangement of tarsi, with a very short triangular 1st tarsomere (basitarsus) and a long 2nd tarsomere ending with simple, paired claws [21]. The basitarsus is extended under the 2nd tarsomere, which is articulated in its dorsal part, suggesting that the 2nd tarsomere moves only mediolaterally relative to the basitarsus [22]. Unlike the tarsi of zorapterans, the tarsi of *F. newi* have two equally long tarsomeres, similarly to those of 1st-instar Dermaptera nymphs [17]. 

Chao and Chen (2000) [9] described and illustrated the mandibles and cerci (Figure 2) of *F. newi*, which allow comparison between Zoraptera and Dermaptera nymphs and classification of this species in Dermaptera. The mandibles are strongly sclerotized, almost triangular with a rounded lateral margin, with well-differentiated molar and incisor parts in both Zoraptera and Dermaptera, but they differ in the number of teeth in the incisor part of the mandible [21]. Five teeth are present on the distal part of the mandibles in Zoraptera (Figure 2C), but only 2 to 3 teeth are present in Dermaptera (Figure 2B) [11,16,22]. In Zoraptera, the left mandible bears stick-like prostheca [11] on the ventral surface of the second (subapical) tooth; in Dermaptera, the prostheca is not developed [23]. The mandible of *F. newi* (Figure 2A) illustrated in the species description by Chao and Chen (2000) [9] undoubtedly ranks this species among the Dermaptera. 

Zorapterans have small, unsegmented cerci inserted laterally on the 9th abdominal tergite. The cerci are as long as or a little longer than wide, are conical with a slightly pointed apex, and are covered with numerous minute spicules and short setae, with one or more long, distally oriented setae [21]. The exception is males of *Spermozoros asymmetricus* (Kočárek, 2017), which have one (right) cercus enlarged with a sickle-shaped, pointed appendage terminating in one large apical tooth [24]. *Formosozoros newi* has long, simply s-shaped cerci that are not covered with spicules, i.e., the cerci of *F. newi* are typical of the cerci of Dermaptera nymphs. Unresolved remains the taxonomy of *Zorotypus longicercatus* Caudell, 1927 from Jamaica (incertae sedis within Zoraptera [24]), which has been described based on nymphs and has similarly shaped cerci as *F. newi* [7,25]. The systematic position of this species will be analyzed in a forthcoming study.

## 5. Conclusions

Based on the morphological comparisons of diagnostic characters, we here identified *Formosozoros newi* as the 1st-instar nymph of an earwig (Dermaptera). A closer classification is not possible, because such classification requires the morphological descriptions of Dermaptera nymphs, which are not available. In addition, the type material of the studied species is lost. We therefore propose that the genus name *Formosozoros* Chao & Chen, 2000 and species name *Formosozoros newi* Chao & Chen, 2000 are nomina dubia.

## Figures and Tables

**Figure 1 insects-14-00053-f001:**
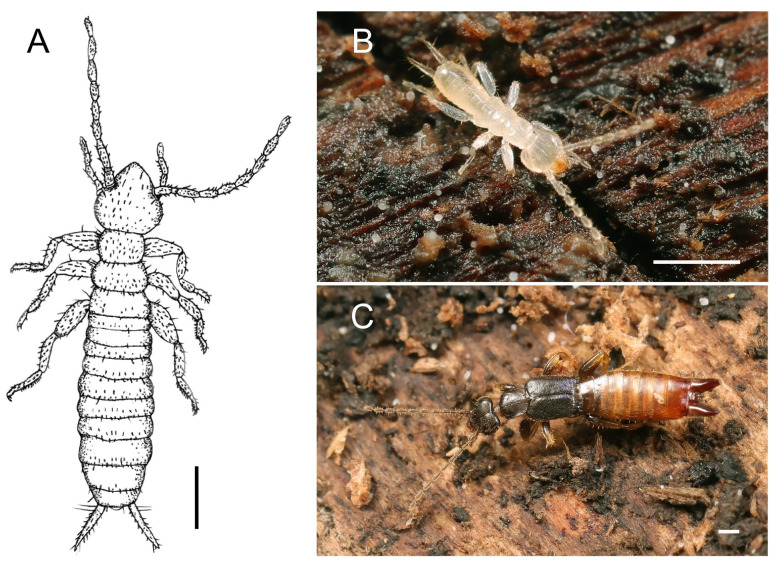
*Formosozoros newi* (Chao & Chen, 2000) (**A**) and *Paralabellula curvicauda* (Motschulsky, 1863) (**B**,**C**). Dorsal view on *F. newi* redrawn from the original description (Chao & Chen 2000) (**A**); 1st-instar nymph of *P. curvicauda* from a breeding culture (**B**); and adult female of *P. curvicauda* from a breeding culture (**C**). Figure A reproduced with permission from Pan-Pacific Entomologist published by the Pacific Coast Entomological Society in 2000. Scale bars = 0.5 mm.

**Figure 2 insects-14-00053-f002:**
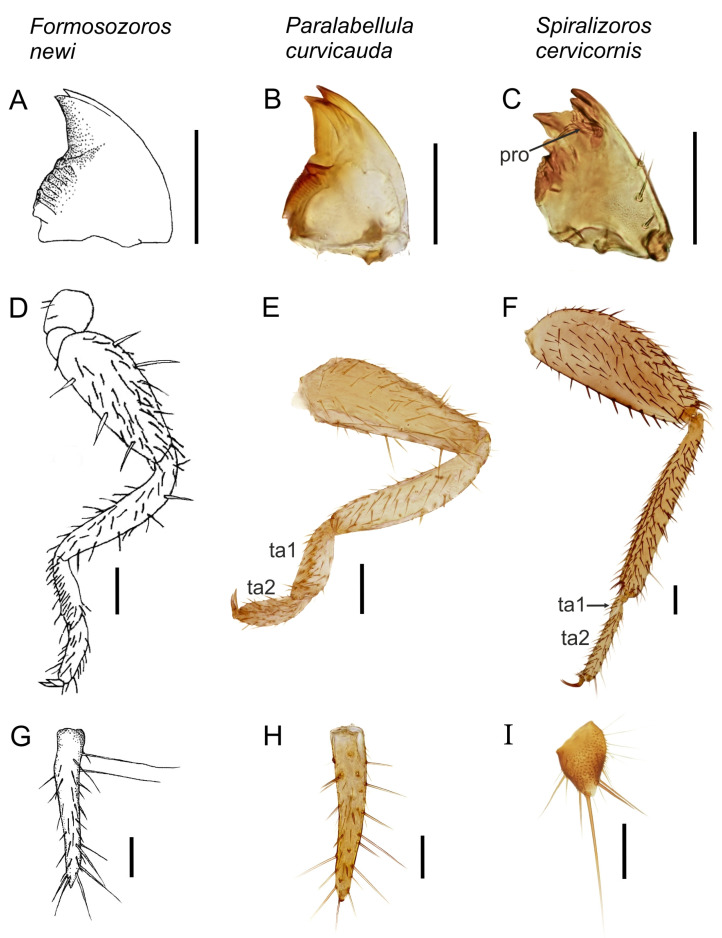
Comparison of diagnostic morphological structures among *Formosozoros newi* (Chao & Chen, 2000) (**A**,**D**,**G**), the nymph of *Paralabellula curvicauda* (Motschulsky, 1863) (**B**,**E**,**H**), and the adult of *Spiralizoros cervicornis* (Mashimo, Yoshizawa & Engel, 2013) (**C**,**F**,**I**). Ventral views of the left mandible of *F. newi* (**A**), *P. curvicauda* (**B**), and *S. cervicornis* (**C**). Dorsal views of the right hind leg of *F. newi* (**D**), *P. curvicauda* (**E**), and *S. cervicornis* (**F**). Dorsal views of the right cercus of *F. newi* (**G**), *P. curvicauda* (**H**), and *S. cervicornis* (**I**). pro—prostheca; ta1—1st tarsomere; ta2—2nd tarsomere. Drawings of *Formosozoros newi* (**A**,**D**,**G**) are redrawn from the original description (Chao & Chen, 2000) with permission from Pan-Pacific Entomologist, published by the Pacific Coast Entomological Society. Scale bars = 0.1 mm.

## Data Availability

The data presented in this study are available in the article.

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
