# Peer review of "An Immature Dermapteran Misidentified as an Adult Zorapteran: The Case of Formosozoros newi Chao & Chen, 2000"

_insects, 2023, doi:10.3390/insects14010053_

Round 1

Reviewer 1 Report

I like this manuscript. The paper "An immature dermapteran misidentified as an adult zorapteran: a case of Formosozoros newi Chao and Chen, 2000" is clear and goes straight to the point. Figures are clear and useful. The text is brief, but it is well structured and from my point of view it demonstrates the misidentification of Formosozoros newi Chao and Chen, 2000, demonstrating that the genus and the species are nomina dubia, actually being a Dermaptera nymph.

Perhaps what worries me the most about the work is that it has not been possible to examine the holotype/paratypes. Nevertheless, it seems that the original material is not available to the scientific community (or at least temporarily unavailable). However, based on the images and data published in the original publication by Chao and Chen (2000), I believe that the authors have more than acceptably evidenced the affinity of F. newi with Dermaptera.

The English is comprehensible, but the MS would ideally benefit from a second correction by a native speaker.

Author Response

We thank for the evaluation of our work!

Reviewer 2 Report

The manuscript entitled “An immature dermapteran misidentified as an adult zorapteran: a case of Formosozoros newi Chao and Chen, 2000” tries to solve the problem of the taxonomic identity of the genus Formosozoros and its monotypic species F. newi.

 The authors support the hypothesis that F. newi is actually the nymph of Dermaptera, therefore considering F. newi as a nomina dubia inside Dermaptera. The arguments put forward by the authors are convincing and this manuscript certainly constitutes an interesting contribution to increase the knowledge of the biodiversity of the Zoraptera. However, the manuscript has a strong taxonomic bias and for this reason I would see it better in a journal more dedicated to animal taxonomy (e.g. Zootaxa).

Regarding the manuscript, it is well written, with a good and comprehensive iconography, however two problems need to be solved before publication:

1)     Both the original description of the species (Chao & Chen, 2000: 26) and the subsequent papers about F. newi (e.g. Engel & Grimaldi, 2000: 155) underline the possible relationship between F. newi and Zorotypus longicercatus. In the present manuscript authors compare F. newi with Spiralizoros cervicornis (I supposed because the latter is a species from SE Asia, but I suggest to detail this point). But no mention to Z. longicercatus has been done by authors. In my opinion it is important to improve the Fig. 2 (the key part of the manuscript) with same figure from Z. longicercatus.

2)     The manuscript is too long, with some repetition. For example in Materials and Methods (Lines 50-52) authors anticipate that types of F. newi have been lost, but this is repetition of Results. In the Results all the part from line 130 to line 143 is not necessary because this is the discussion about the results and in fact it is repeated in this last part. Also the detailed description of F. newi is not necessary since it is yet present in Cheao & Chen, 2000: I suggest to eliminate this part, eventually reporting only key features which help in separating F. newi from other Zorapteran species.

I found the manuscript well written with only few typos errors that I underlined directly in the attached files.

Author Response

Comment 1: We are aware of the similarity of F. newi with Z. longicercatus and we will address the taxonomic position of this species in the near future. Our original intention was to solve the taxonomic positions of both species at the same time, but unfortunatelly the type specimens of Z. longicercatus are temporarily unavailable for the study (communication with dr. Floyd Shockley, the currator in The National Museum of Natural History in Washington). According to the drawings and the original description we suppose, that this species is also Dermaptera nymph, but because the type material exist, we feel obliged to study this material first and don't jump to conclusions only according to the description which is not very detailed. However, based on the comment, we have added information about the similar long cerci of Z. longicercatus and that we will focus on the taxonomy of this species.

Comment 2: We thank the reviewer for all valuable comments on the structure of the manuscript and for pointing out repetitive parts. We edited the manuscript according to the reviewer's suggestions (including comments in the text of the manuscript), we shortened the descriptive parts and omitted repetitive information about the loss of type material. However, we do not agree with moving the content of subsection 3.3 to the Discussion because, unlike the reviewer, we perceive this subsection as a crucial part of the results, where taxonomic changes are proposed and these taxonomic changes are justified here. In our opinion, it is not a polemic with the results, but the results themselves.